# Influence of the Maillard Reaction on the Properties of Gelatin/Zein Nanofibers Loaded with Dihydromyricetin Prepared by Electro-Blowing Spinning

**DOI:** 10.3390/biom15060891

**Published:** 2025-06-18

**Authors:** Hui Xiang, Runtian Wu, Man Xiao, Jianhui An, Longchen Shang, Yexing Tao, Lingli Deng

**Affiliations:** 1Hubei Key Laboratory of Biological Resources Protection and Utilization, Hubei Minzu University, Enshi 445002, China; 202430415@hbmzu.edu.cn; 2College of Biological and Food Engineering, Hubei Minzu University, Enshi 445002, China; 202311603@hbmzu.edu.cn (R.W.); 202212820@hbmzu.edu.cn (M.X.); 2020049@hbmzu.edu.cn (J.A.); 2021021@hbmzu.edu.cn (L.S.)

**Keywords:** zein, gelatin, dihydromyricetin, Maillard reaction, antioxidant activity

## Abstract

This study investigated gelatin/zein nanofibers loaded with dihydromyricetin (0–20%, relative to protein weight), before and after the Maillard reaction (60 °C with 50% relative humidity for 6 h). Scanning electron microscopy and diameter distribution analysis indicated that dihydromyricetin incorporation increased the fiber diameter from 692 ± 133 to 922 ± 121 nm, while the nanofibers maintained a uniform morphology following the Maillard reaction. Fourier transform infrared spectroscopy revealed that dihydromyricetin formed hydrogen bonds with protein molecules. X-ray diffraction results indicate that dihydromyricetin was uniformly dispersed within the gelatin/zein nanofibers. The addition of dihydromyricetin improved the thermal stability of the nanofibers. Furthermore, after the Maillard reaction, the nanofibers with dihydromyricetin demonstrated enhanced water resistance. Mechanical testing revealed that nanofibers containing 20% dihydromyricetin after the Maillard reaction exhibited a considerably higher elastic modulus of approximately 90 MPa. In addition, nanofibers containing dihydromyricetin exhibited notable antioxidant activity and antibacterial properties against *Escherichia coli* and *Staphylococcus aureus*. In summary, gelatin/zein nanofibers containing high concentrations of dihydromyricetin exhibited favorable physical and functional properties, supporting their suitability as effective delivery systems for dihydromyricetin in active packaging applications.

## 1. Introduction

Dihydromyricetin is the most abundant and effective component among the flavonoids of *Ampelopsis grossedentata*, possessing antioxidant properties, antibacterial activity, anti-inflammatory capabilities, anticancer effects, liver-protective functions, and neuroprotective benefits [1,2]. However, despite its promising therapeutic potential, its practical application faces considerable challenges, primarily due to its limited water solubility and low bioavailability. To address these limitations, researchers explored various encapsulation strategies. Including nanocapsules [3], nanoemulsions [4], microemulsions [5], hydrogels [6], and nanofibers [7], which enhanced the solubility and bioavailability of dihydromyricetin. Among these delivery systems, nanofibers emerged as a novel method for the controlled delivery of bioactive compounds while preserving their bioactivity [8]. Ju et al. [9] reported that the high surface-to-volume ratio of nanofibers considerably improves encapsulation efficiency and bioactivity. Consequently, nanofibers incorporating bioactive compounds, such as dihydromyricetin, are primarily used in active food packaging applications, effectively providing antimicrobial and antioxidant benefits that enhance food preservation and safety [10]. A previous study developed dextran/zein/xylose nanofibers containing 0–4% dihydromyricetin, with the dextran/zein matrix serving as an effective delivery vehicle for this bioactive compound and exhibiting tunable physical properties [11]. However, its low loading capacity limits its effectiveness in long-term sustained release and high-dose therapeutic applications. To address these limitations, high-concentration encapsulation is essential to enhance the therapeutic efficacy and bioavailability of dihydromyricetin in various applications [12]. However, there is limited research on the encapsulation of high concentrations of dihydromyricetin within nanofibers.

Electrospinning technology has various applications, including the delivery and controlled release of active compounds [13], encapsulation of probiotics [14], development of biosensors [15], removal of heavy metals [16], and production of active food packaging materials [17,18]. Electro-blowing spinning is a hybrid manufacturing technique that integrates the powerful forces of an electric field with high-pressure gas, overcoming limitations such as low productivity in electrospinning and jet instability in solution blow spinning [19]. Although researchers primarily focused on electrospinning using synthetic polymers, in recent years, natural polymers received greater attention due to the increasing emphasis on environmental sustainability [20]. This interest in natural polymers stems from their inherent safety, biocompatibility, and biodegradability [21]. Zein, a plant-derived protein, exhibits excellent biodegradability, biocompatibility, and edibility, while notably enhancing the hydrophobic properties of composite films. Gelatin, a versatile biopolymer derived from collagen, also offers excellent biodegradability, emulsification capacity, biocompatibility, and high viscosity, making it highly applicable in the food industry [22].

Numerous researchers fabricated nanofibers using gelatin, zein, or a mixture of the two. Zhang et al. [23] successfully prepared gelatin nanofibers, while Zhang et al. [24] found that zein-based nanofibers exhibited poor mechanical properties and solvent resistance. To address these limitations, improvements in the water/oxygen barrier performance and mechanical characteristics of the films are necessary. Utilizing the complex physical and chemical interactions between polymer chains can substantially enhance the physicochemical characteristics of materials derived from biopolymers [25]. This approach is particularly effective when integrated with the Maillard reaction [26]. The Maillard reaction, which forms covalent bonds between reducing sugars and proteins, attracted widespread interest because it can improve both the physical and chemical properties [27]. However, despite its broad reaction conditions, it may degrade the functional properties of bioactive compounds under high-temperature conditions. Kwak et al. [28] demonstrated that the Maillard reaction occurred under elevated temperature conditions, leading to functional deterioration and the generation of unwanted by-products. Previous studies successfully synthesized gelatin/zein nanofibers, which underwent effective crosslinking at 60 °C and 50% relative humidity [29]. Following crosslinking with glucose, the fibers demonstrated considerably enhanced water resistance and the formation of a more rigid and reinforced network structure.

To overcome the solubility limitations of dihydromyricetin, this study focused on fabricating gelatin/zein nanofibers containing dihydromyricetin at concentrations ranging from 0% to 20%. Crosslinking was performed under mild conditions using the Maillard reaction. The morphology and diameter distribution of the nanofibers were assessed by scanning electron microscopy (SEM). Analytical techniques, including Fourier transform infrared spectroscopy (FTIR), X-ray diffraction (XRD), thermal analysis, water contact angle (WCA) measurements, and X-ray photoelectron spectroscopy (XPS), were utilized to examine the interaction between protein molecules and dihydromyricetin. Furthermore, the mechanical characteristics and water vapor permeability (WVP) of the nanofibers were examined to evaluate their suitability for food packaging applications. The antioxidant and antibacterial capabilities of the dihydromyricetin-incorporated gelatin/zein nanofibers were also investigated.

## 2. Materials and Methods

### 2.1. Chemicals

Gelatin, glucose, and dihydromyricetin were obtained from Aladdin Reagent Database Inc. (Shanghai, China). Zein (Z3625), 2,2 diphenyl-1-picrylhydrazyl (DPPH), 2,2′-azino-bis (3-ethylbenzothiazoline-6-sulfonic acid) (ABTS), and 2,4,6-tri (2-pyridinyl)-1,3,5-triazine (TPTZ) were obtained from Sigma Aldrich (St. Louis, MO, USA). Acetic acid and some other reagents were supplied by the National Pharmaceutical Group Chemical Reagent Co., Ltd. (Shanghai, China).

### 2.2. Liquids for Electrospinning

Electrospinning liquids were prepared as follows: 1.5 g of gelatin and 1.5 g of zein were dissolved in 10 mL of 80% acetic acid solution (at an 8:2 volume ratio of acetic acid to water). Then, 0.5 g of glucose was introduced into the mixture based on a previous study [29]. Dihydromyricetin was incorporated into the solution at concentrations of 0%, 5%, 10%, 15%, and 20% based on the protein weight. The samples were continuously stirred overnight to ensure thorough dissolution.

### 2.3. Fiber Fabrication Process

As displayed in Figure 1, the electro-blowing spinning apparatus included an air compressor, a high-voltage power supply, a syringe pump, a grounded stainless steel rotating drum, and a 10 mL plastic syringe. The spinning parameters were set as follows: a syringe pump flow rate of 10 mL/h, an applied voltage of 18 kV, a distance of 15 cm from the needle tip to the drum collector, and an airflow rate of 400 L/h maintained by an air pump. The resulting nanofibers, incorporating dihydromyricetin at concentrations of 0%, 5%, 10%, 15%, and 20% relative to protein weight, were labeled as D0, D5, D10, D15, and D20, respectively.

### 2.4. Maillard Reaction

The Maillard reaction was performed in a controlled environment chamber (HSX-150L, Shanghai Gipp Electronic Technology Co., Ltd., Shanghai, China) at 60 °C with 50% relative humidity for 6 h. The resulting nanofibers, which incorporated dihydromyricetin at concentrations of 0%, 5%, 10%, 15%, and 20% relative to protein weight, were labeled as D0M6, D5M6, D10M6, D15M6, and D20M6, respectively.

### 2.5. Characterization of the Fiber Properties

The morphology of the nanofibers was examined using SEM (Gemini SEM 300, ZEISS, Oberkochen, Germany). Prior to measurement, the samples underwent vacuum gold sputter coating to enhance conductivity and minimize charging effects. The accelerating voltage was set to 3 kV, with the magnification maintained at 1000×. The diameters of the fibers were measured by randomly selecting 40 fibers from each sample and analyzing them with Nano Measurer 1.2 software; the fiber diameter distribution was then evaluated. To further investigate the structural properties of the nanofibers, FTIR spectra were collected using an FTIR spectrometer (Thermo Fisher Scientific, Inc., Waltham, MA, USA) over the range of 4000–400 cm^−1^ at a spectral resolution of 4 cm^−1^, obtaining 32 scans per measurement [30]. The resulting spectral peaks were analyzed using OMNIC 8.2 and Origin software 2025. For crystal structures analysis, XRD (Bruker-AXS GmbH, Karlsruhe, Germany) with a Cu target was employed, and diffraction data were collected over a 2θ range of 5–90° at a scanning speed of 1° min^−1^, operating at 40 kV and 200 mA [31].

Differential scanning calorimetry (DSC) and thermogravimetric analysis (TGA) were employed to evaluate the thermal characteristics of the fabricated nanofibers, utilizing a simultaneous thermal analyzer from (NETZSCH-Gerätebau GmbH, Selb, Germany). For each sample, approximately 6–10 mg was measured and placed into an aluminum crucible. The heating process was conducted from 30 °C to 600 °C at a rate of 10 °C/min under a nitrogen atmosphere. An empty crucible served as the control under identical conditions. The WCAs of the prepared nanofibers were measured using a contact angle measurement system (OCA-20, Dataphysics Instruments, Filderstadt, Germany) following the method described by Jiang et al. [32]. The nanofibers were fixed on glass slides, and deionized water served as the probe liquid. Measurements were performed at ambient temperature, with each sample evaluated in triplicate.

XPS analysis employing an AXIS SUPRA apparatus (Kratos Analytical Inc., Manchester, UK) was performed to evaluate the surface chemical composition of the nanofibers. Full-spectrum scans were performed over a binding energy range of 0–1350 eV with a pass energy of 100 eV, while high-resolution C1s spectra were acquired at a pass energy of 50 eV [30]. To determine WVP, a container was filled with 10 mL of deionized water, and its edge was then sealed with nanofibers with a diameter of 6 cm and uniform thickness. The initial mass was recorded, and the container was placed inside a desiccator. Sample masses were recorded at hourly intervals for a total of 6 h. WVP was then determined using the following equation [33]:$$\mathrm{WVP}\left(g/m\cdot s\cdot \mathrm{pa}\right)=\frac{W_{s}}{\mathrm{At}}\times \frac{L}{∆P}$$ where A represents the contact area between the nanofibers and water vapor (cm^2^), L signifies the thickness (cm), ∆P denotes the applied vapor pressure difference (Pa) (2237.8 Pa at 28 °C), and Ws/t corresponds to the linear regression of weight change over time (g/s). Each sample was evaluated in triplicate.

To assess the mechanical performance of the nanofibers, specifically the tensile strength (TS), elastic modulus (EM), and elongation at break (EB), a DR-508A automated tensile testing machine (Dongri Instrument Ltd., Dongguan, China) was employed. Nanofiber samples were trimmed into 10 × 3 mm rectangular strips and then fixed on the testing apparatus. Before conducting the tests, the dimensions (length, width, and thickness) of each specimen were measured for EM calculation. The specimens were stretched with a force of 5 N at a loading rate of 5 mm/min. Each specimen underwent five replicate tests to ensure consistent results. TS, EM, and EB were determined using methods established in previous research [34].

### 2.6. Antioxidant and Antimicrobial Activities

To evaluate the antioxidant capacity, the free radical scavenging rate, along with Fe^3+^ and Cu^2+^ reduction methods, were employed following previously established methodologies [35,36]. Each experiment was performed in triplicate. In the DPPH radical scavenging assay, 1 mg of nanofibrous film containing varying concentrations of dihydromyricetin was transferred into a centrifuge tube containing 2 mL of DPPH solution. The mixture was incubated in the dark for 30 min, followed by a measurement of absorbance at 517 nm. The proportion of free radical scavenging activity was assessed using to the following equation: $$\mathrm{DPPH}\;\mathrm{radical}\;\mathrm{scavenging}\;\mathrm{rate}\;(\%)=\frac{A_{0}-A_{1}}{A_{0}}\times 100$$ where A_0_ and A_1_ correspond to the absorbance values for the control and test samples, respectively.

To prepare the ABTS working solution, ABTS (0.0384 g) and K_2_S_2_O_8_ (0.0134 g) were each dissolved in 10 mL of deionized water. Equal volumes of the two solutions were mixed and allowed to react in the dark for 12 h. The resulting solution was diluted with PBS buffer until the absorbance at 734 nm was adjusted to 0.7 ± 0.02. For analysis, 1 mg of nanofibrous film with various concentrations of dihydromyricetin was added to 2 mL of the ABTS/PBS solution. After the mixture was incubated in the dark for 30 min, the absorbance was measured at 734 nm. The proportion of free radical scavenging activity was determined using the following equation:$$\mathrm{ABTS}\;\mathrm{radical}\;\mathrm{scavenging}\;\mathrm{rate}\;(\%)=\frac{A_{0}-A_{1}}{A_{0}}\times \;100$$ where A_0_ denotes the absorbance of the blank, whereas A_1_ corresponds to the absorbance of the sample.

To prepare the TPTZ stock solution, 0.03127 g of TPTZ was dissolved in 10 mL of HCl at a concentration of 40 mmol/L. Subsequently, the TPTZ working solution was prepared by combining a 20 mmol/L FeCl_3_ solution, the TPTZ stock solution, and a 0.3 mol/L CH_3_COONa buffer in a volume ratio of 1:1:10. For the TPTZ assay, 1 mg of nanofibrous film with various concentrations of dihydromyricetin was added to 2 mL of the TPTZ working solution and incubated at 37 °C for 30 min. Subsequently, the absorbance was determined at a wavelength of 592 nm using a spectrophotometer. The reducing ability was evaluated based on the optical density (OD) value, with each measurement performed in triplicate for accuracy.

To prepare the working solution for the metal ion reduction assay, a solution with a concentration of 0.01 mol/L Cu^2^⁺, a solution containing 1 mol/L of CH_3_COONH_4_, and a solution containing 7.5 mmol/L of C_14_H_12_N_2_ were combined in equal volumes. For the assay, 1 mg of nanofibrous film with various concentrations of dihydromyricetin was added to 2 mL of the working solution. The mixture was incubated at 25 °C for 30 min, after which the absorbance was measured at 450 nm using a spectrophotometer. The reducing ability of the sample was determined based on the OD value at this wavelength, with each experiment performed in triplicate to ensure reliability of the results.

To assess the antibacterial properties of the nanofibers, the inhibition zone assay was employed, with *Escherichia coli* (*E. coli*) and *Staphylococcus aureus* (*S. aureus*) bacteria serving as the test strains. The process included cutting the nanofibrous film into 5 mm diameter discs, followed by sterilization under ultraviolet light for 30 min. Subsequently, 100 µL of bacterial solution (1 × 10^6^ CFU/mL) was uniformly spread on agar Petri dishes, and the sterilized fiber discs were placed on the agar surface. After 24 h of incubation at 37 °C, the size of the clear zones where bacterial growth was inhibited was determined using a precision caliper. Each test was repeated in triplicate to ensure the consistency of the results.

All test results are presented as mean values accompanied by their standard deviations, derived from at least three independent replicates. Statistical evaluations were conducted employing one-way analysis of variance, complemented by Tukey’s post-hoc analysis to determine significance (*p* < 0.05). The results were then visualized using Origin Pro 2024b (OriginLab Corporation, Northampton, MA, USA).

## 3. Results and Discussion

### 3.1. Morphology of Nanofibers

SEM images and diameter distributions of the gelatin/zein nanofibers with various dihydromyricetin ratios, before and after a 6 h Maillard reaction, are presented in Figure 2. As the dihydromyricetin concentrations increased from 0% to 20%, the mean diameter of the nanofibers increased considerably from 692 ± 133 nm (D0) to 922 ± 121 nm (D20). However, the addition of dihydromyricetin did not substantially affect the uniform and round morphology of the fibers. It is well known that the diameter distribution of electrospun nanofibers is affected by polymer concentration [37]. The change in fiber diameter may be due to the fact that incorporating bioactive compounds alters the viscosity of the electrospinning solution [38]. Liu et al. [7] also observed that incorporating dihydromyricetin resulted in an increase in the diameter of polyvinylpyrrolidone/chitosan nanofibers. Increasing the polymer concentration promotes increased entanglement of polymer chains, which in turn increases viscosity [39]. A suitable level of entanglement facilitates the formation and morphology of films. The increased viscosity of the solution affected the electrospinning process by hindering the splitting of droplets at the tip of the needle, consequently increasing the diameter of the resulting nanofibers.

A comparison of the fiber morphology before and after the Maillard reaction revealed that the diameter of the nanofibers remained unchanged. The addition of dihydromyricetin did not affect the uniform and smooth appearance of the nanofibers, suggesting that the structure of the nanofibers remained stable under the applied heat treatment conditions. The preservation of the structural integrity and high surface-to-volume ratio of the nanofibers after crosslinking considerably improved their effectiveness as advanced carriers for bioactive compounds.

### 3.2. FTIR Spectra

Figure 3 displays the FTIR spectra of gelatin/zein film composites containing various concentrations of dihydromyricetin before (Figure 3a) and after (Figure 3b) undergoing the Maillard reactions for 6 h. The absorption peaks between 3200 and 3300 cm^−1^ represent O–H and N–H stretching vibrations, which likely indicate hydrogen bonding interactions between gelatin and zein [38]. The absorption bands between 2800 and 2900 cm^−1^ are indicative of the asymmetric stretching vibrations of C–H bonds. The peaks detected at approximately 1637–1647 cm^−1^, 1539 cm^−1^, and 1243–1247 cm^−1^ correspond to C=O stretching (amide I), C–N stretching (amide II), and a combination of C–N stretching and N–H bending (amide III), respectively. The peak observed near 1454 cm^−1^ in the nanofibers is associated with the stretching vibrations of the C–N and N–H bonds, while the peak near 1080 cm^−1^ is related to C–H bending and C–O stretching vibrations [31]. As the concentration of dihydromyricetin increased, an enhancement in peak intensity at approximately 1170 cm^−1^ was observed, which is attributed to the C–O stretching vibrations of phenolic hydroxyl groups present in dihydromyricetin [40]. This finding provides strong evidence of the successful loading of dihydromyricetin at elevated concentrations.

The absence of significant shifts in amide I and III after crosslinking suggests that the mild Maillard reaction conditions and short reaction time were insufficient to alter the protein structure. Zhang et al. [41] subjected gluten/zein fiber membranes to the Maillard reaction under mild conditions at 60 °C and 40% relative humidity for 24 h, observing only minor shifts in the amide bands. In contrast, Ahmed et al. [42] subjected gelatin fiber membranes to the Maillard reaction at a high temperature of 140 °C, resulting in more pronounced shifts in the amide bands.

### 3.3. XRD Spectra

Figure 4 presents the XRD spectra of gelatin/zein nanofibrous films containing various concentrations of dihydromyricetin before (Figure 4a) and after (Figure 4b) the Maillard reaction for 6 h. The gelatin/zein nanofibrous film exhibited a narrow peak at 8.38° and a broad peak at 21.03°. Ki et al. [43] reported that the reduced crystallinity in the majority of gelatin/zein nanofibers may be due to acidic degradation affecting their structural order. Dihydromyricetin exhibits distinct diffraction peaks in the range of 10° to 30°, indicating its crystallinity [44]. In this study, consistent diffraction patterns were detected across all samples. With the increase in the concentration of dihydromyricetin from 5% to 20%, the nanofibers displayed a narrow peak at approximately 9° and a broad peak at approximately 21°. This suggests low crystallinity and the absence of new characteristic peaks, indicating that the electrospinning process may hinder polymer crystallization [45]. These results indicate favorable compatibility and synergistic effects between gelatin, zein, and dihydromyricetin within the nanofiber films. Chen et al. [46] reported that the diffraction patterns of proteins feature peaks around 9°and 21° associated with α-helix and β-sheet conformations, respectively.

Following the Maillard reaction, the decreased intensity of the diffraction peaks indicated a reduction in the crystallinity of the nanofibers. Sun et al. [47] reported the relationship between the XRD pattern and protein secondary structure. Protein secondary structure refers to the local folding patterns within the protein molecule, including α-helices, β-sheets, β-turns, and random coils, which are maintained by non-covalent interactions such as hydrogen bonds and van der Waals. Pan et al. [48] suggested that the Maillard reaction involves condensation between carbonyl groups and ε-amino groups, resulting in the reduced α-helix content. The reduced crystallinity of the nanofibers may therefore be due to damage to the protein secondary structure. Another reason may be that glucose conjugated with protein enhanced the mobility of protein polymer chains, thereby reducing the crystalline structures by disrupting the arrangement of protein chains [49].

### 3.4. Thermal Characteristics

Figure 5a–c displays the DSC, TGA, and derivative thermogravimetry curves, respectively, of nanofibers loaded with varying concentrations of dihydromyricetin, while Figure 6 illustrates the nanofibers after undergoing the Maillard reaction for 6 h. The DSC and TGA results are summarized in Table 1. The distinct thermal absorption peaks and their corresponding enthalpy values observed in the DSC curves correspond to the denaturation temperature and the enthalpy change associated with denaturation, respectively. The denaturation temperature increased notably from 70.7 °C to 82.4 °C with the addition of dihydromyricetin. This temperature increase suggests that the hydrogen bonds between dihydromyricetin and the protein molecules in the nanofibers enhanced thermal stability. The FTIR and XRD results also indicate that hydrogen bonds existed between dihydromyricetin and the polymer. Wang et al. [50] reported similar findings, demonstrating that incorporating perillaldehyde into nanofiber films enhanced their thermal stability. Upon completion of the Maillard reaction for 6 h, the denaturation temperatures of D0M6, D5M6, D10M6, D15M6, and D20M6 were 66.7 °C, 64.2 °C, 64.1 °C, 65.6 °C, and 63.9 °C, respectively.

The degradation of nanofibers was characterized by three distinct phases. The first peak indicated the evaporation of absorbed moisture, while the second peak, observed at approximately 200 °C, was associated with protein decomposition. A similar result was reported by Deng [51], who developed the encapsulation of allopurinol using glucose-crosslinked gelatin/zein nanofibers. The most substantial weight loss was observed in the third stage at approximately 320 °C, which corresponded to fiber carbonization. Notably, after the Maillard reaction, the nanofibers displayed reduced degradation temperatures, indicating reduced thermal stability. This phenomenon may be attributed to the intermolecular interactions and diminished crystallinity consistent with the XRD results.

### 3.5. WCA

The WCA of the fibers is presented in Figure 7, illustrated by water drop profiles. It is widely known that a higher WCA typically indicates the increased surface hydrophobicity of fibers [52]. The un-crosslinked films exhibited hydrophobic properties, with the contact angle increasing from 107.9 ± 0.5° to 132.2 ± 0.1° as the concentration of dihydromyricetin increased from 5% to 20%. This phenomenon can be attributed to the incorporation of dihydromyricetin, which enhanced hydrogen bonding interactions with gelatin/zein while also contributing its own hydrophobic characteristics. Deng et al. [53] reported that the interaction between gelatin and zein molecules, facilitated by hydrogen bonds from their hydrophilic groups, contributed to the formation of a more hydrophobic exterior surface. Moreover, the addition of thymol was found to increase hydrogen bonding interactions of gelatin/zein films [38].

After undergoing the Maillard reaction for 6 h, the WCAs of D0M6, D5M6, D10M6, D15M6, and D20M6 were 123.6°, 137.3°, 151.7°, 140.8°, and 143.3°, respectively, indicating increased hydrophobicity after crosslinking. A previous study also reported that the Maillard reaction occurs between the aldehyde group of glucose and the amino groups of proteins, leading to the orientation of less polar groups outward and consequently yielding a more hydrophobic surface [54].

### 3.6. Surface Elemental Analysis

XPS was employed to determine the surface elemental composition of the nanofibers, with the results presented in Figure 8. The N atomic content decreased considerably as the concentration of dihydromyricetin increased from 0% to 20%. The decreased N atomic content may be explained by the nature of dihydromyricetin as a flavonoid compound. When co-spinning with gelatin/zein, dihydromyricetin is distributed in small amounts on the nanofiber surface. The C and O atomic proportions exhibited no significant change. Figure 9 displays the high-resolution C1s spectra, which reveal three notable peaks at 284.7 eV, 286.2 eV, and 288.0 eV, corresponding to C–C/C=C, C–O/C–N, and O–C=O bonds, respectively [55]. The proportion of C–C/C=C bonds decreased from 61.51% to 60.10%, while the proportion of C–O/C–N bonds increased from 17.60% to 20.31%.

Following a 6 h Maillard reaction, the C atomic ratio increased, while the O and N atomic ratios decreased. These changes in atomic composition may be attributed to the extent of crosslinking and can be interpreted as indicators of the start of the Maillard reaction [56]. The reduction in O atomic content is likely due to water loss during the crosslinking process. Meanwhile, the reduction in N atomic content may be attributed to the reaction between carbonyl and amino groups, resulting in the formation of volatile by-products; similar findings have been reported by Lee et al. [28]. As the proportion of O and N atomic content decreased, the hydrophobicity of the nanofibers increased. The proportion of C–C/C=C bonds decreased from 63.12% to 60.91%, while the proportion of C–O/C–N bonds increased from 15.36% to 18.15%.

### 3.7. WVP

The moisture barrier capability of nanofibers and the characteristics of active packaging systems are highly influenced by WVP. Figure 10 illustrates the WVP of nanofibers incorporating varying concentrations of dihydromyricetin. As is well known, the WVP of a film is influenced by several factors, including its crystallinity, the balance between hydrophobic and hydrophilic properties, the presence of water vapor channels, and the chemical structure of the polymers [57]. In this study, the incorporation of dihydromyricetin did not notably influence the WVP, except for a marked increase at D20.

After the Maillard reaction, the WVP exhibited had an upward trend. This phenomenon may be attributed to the enhanced water stability resulting from the Maillard reaction, which preserves the porous structure during water vapor diffusion [58]. A previous study reported that materials with high WVP are ideal for packaging fresh foods, whereas materials with low WVP are more suitable for packaging processed foods [59]. Therefore, crosslinked dihydromyricetin-loaded gelatin/zein films demonstrate potential for fresh food packaging applications.

### 3.8. Mechanical Characteristics

Figure 11a–c presents the mechanical characteristic of nanofibers containing various concentrations of dihydromyricetin, specifically, TS, EB, and EM, respectively. Mechanical characteristics play a critical role in active packaging and are affected by the type of polymer base and bioactive substance, biocompatibility, concentration levels, and parameters of the electrospinning process [60]. In this study, the TS and EB of the films decreased with increased dihydromyricetin concentration, whereas the EM increased. These results may be associated with the variations in fiber diameter induced by dihydromyricetin, which are governed by polymer chain interactions. As reported in a previous study, the addition of dihydromyricetin resulted in a reduction in the EB of konjac glucomannan/gellan gum films [61]. In addition, Wang et al. [1] found that incorporating dihydromyricetin into PCL nanofibers at levels of 2%, 6%, and 10% resulted in an increase in the EM and a subsequent decrease in the EB.

Following the Maillard reaction, gelatin/zein nanofibrous film without dihydromyricetin exhibited a substantial increase in EM, which may be due to the formation of compact structures through protein–glucose crosslinking and the enhancement of both intermolecular chain entanglements and interactions within the polymer network [58,62]. However, the nanofibrous film containing 20% dihydromyricetin after the Maillard reaction exhibited the highest EM, which was ascribed to the role of dihydromyricetin in promoting the formation of crosslinked networks. The formation of these crosslinked networks is likely limited the mobility of polymer chains, which was indicated by the reduced EB for D5M6, D10M6, D15M6, and D20M6.

### 3.9. Antioxidant Activities

Figure 12 illustrates the antioxidant activities of gelatin/zein nanofibers, including free radical scavenging ability (DPPH-a, ABTS-b) and ionic reducing capacity (Fe^3+^-c, Cu^2+^-d). The antioxidant properties of gelatin-derived peptides, combined with the reducing capacity of amino acid residues, short peptides, and lutein in zein, considerably influenced the free radical scavenging efficiency and the reduction in Fe^3+^ and Cu^2+^ in unloaded dihydromyricetin films [60,63]. With the addition of dihydromyricetin, the free radical scavenging activity and reducing power increased considerably, which is consistent with a previous study.

In comparison to the un-crosslinked nanofibers, the DPPH free radical scavenging activity demonstrated a substantial decrease following the Maillard reaction. In addition, the Fe^3+^-reducing power of the D5M6 and D10M6 samples and the Cu^2+^-reducing powers of the D10M6 and D15M6 samples exhibited a substantial decrease. These phenomena can be attributed to the increased hydrophobicity of the fibers and the enhanced complexity of the fiber pathways, resulting in a reduced release rate.

### 3.10. Antibacterial Activities

Figure 13 illustrates the antibacterial properties of nanofibers containing varying concentrations of dihydromyricetin, as indicated by the inhibition areas targeting *E. coli* and *S. aureus*. All fibers containing dihydromyricetin exhibited antibacterial activity, suggesting that the encapsulated compound was effectively released. It exhibits significant inhibitory effects against both *E. coli* and *S. aureus*. Compared to the large intestine, Staphylococcus aureus shows no significant changes in concentration before and after the reaction. This discrepancy was likely due to the thicker cell wall of *S. aureus*, which consists of multiple peptidoglycan layers and restricts the entry of hydrophobic compounds [64]. As a flavonoid, dihydromyricetin interferes with bacterial growth and metabolism by damaging the structure of bacterial cell walls and membranes and disrupting metabolic processes [65]. As the dihydromyricetin concentration increased from 5% to 15%, the antibacterial properties of the nanofibers against *E. coli* decreased both before and after crosslinking. This reduction may be attributed to the increased hydrophobicity of the nanofiber membrane surface, a result of dihydromyricetin incorporation and the subsequent Maillard reaction promoting crosslinking. The altered surface properties subsequently hindered the diffusion of dihydromyricetin into the culture medium. At a dihydromyricetin concentration of 20%, the ability of the crosslinked nanofibers to retain dihydromyricetin decreased, resulting in a greater amount of dihydromyricetin released from the fiber membrane surface.

## 4. Conclusions

In this study, gelatin/zein nanofibers containing high concentrations of dihydromyricetin were successfully fabricated using the electro-blowing spinning process. Subsequently, the dihydromyricetin-loaded fibers were subjected to crosslinking through the Maillard reaction. The components (dihydromyricetin, gelatin, and zein) exhibited remarkable compatibility. The incorporation of dihydromyricetin considerably improved the thermal stability, hydrophobicity, and flexibility of the nanofibers, while maintaining the antibacterial and antioxidant capacity. After the Maillard reaction, the nanofiber components were uniformly distributed, exhibiting superior hydrophobicity and increased EM. Overall, the gelatin/zein nanofibers containing high levels of dihydromyricetin exhibited excellent physical properties and bioactivity, demonstrating considerable potential as innovative active food packaging materials.

## Figures and Tables

**Figure 1 biomolecules-15-00891-f001:**
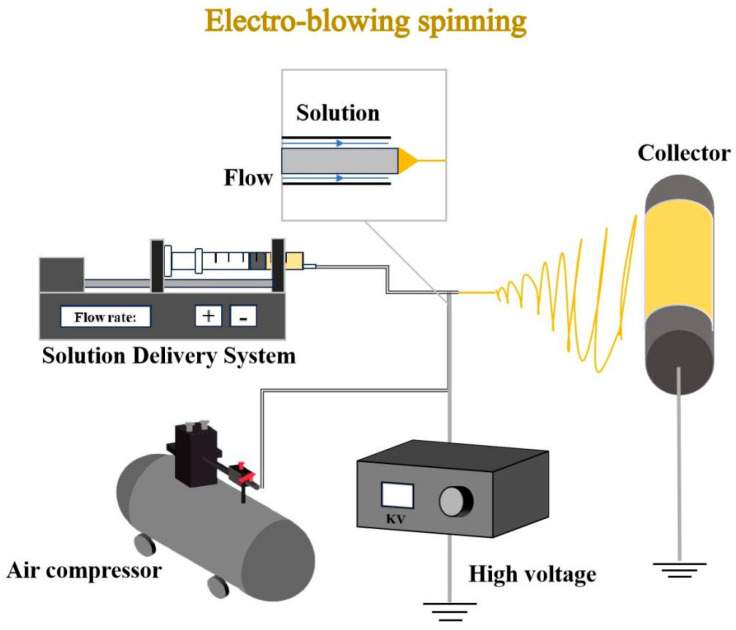
Schematic diagram of electro-blowing spinning equipment.

**Figure 2 biomolecules-15-00891-f002:**
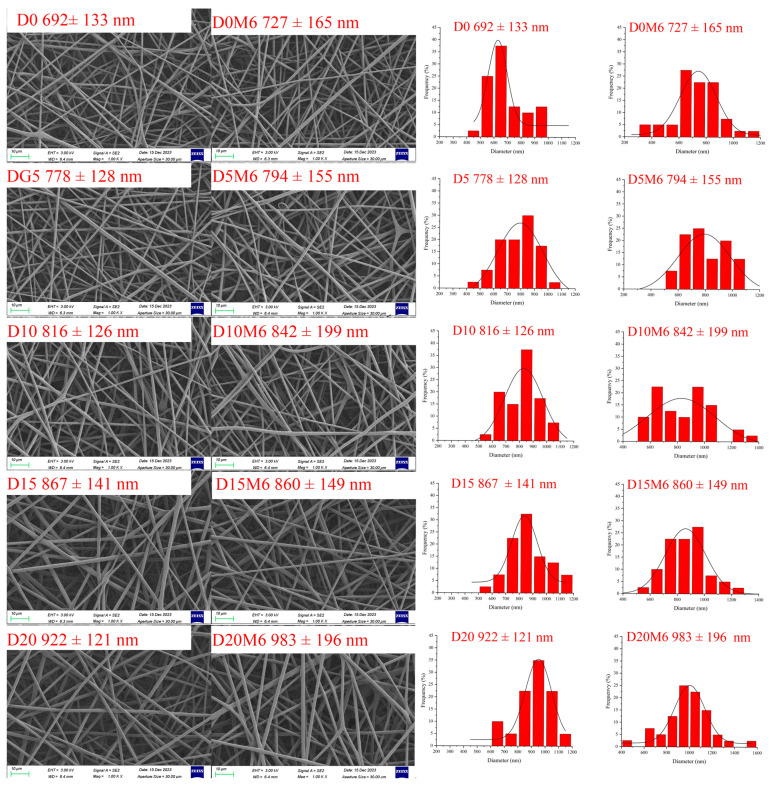
SEM images and fiber diameter distribution of gelatin/zein nanofibers loaded with 0%, 5%, 10%, 15%, and 20% dihydromyricetin before and after undergoing the Maillard reaction for 6 h.

**Figure 3 biomolecules-15-00891-f003:**
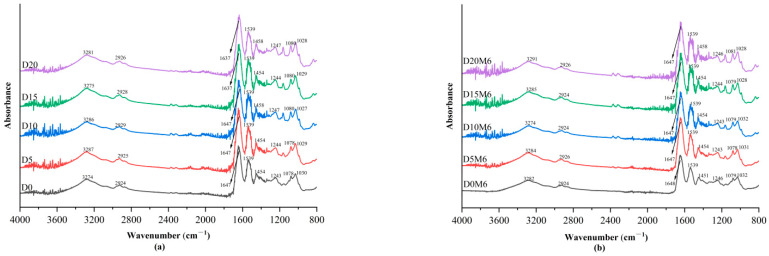
FTIR spectra of gelatin/zein nanofibers loaded with 0%, 5%, 10%, 15%, and 20% dihydromyricetin (**a**) before and (**b**) after undergoing the Maillard reaction for 6 h.

**Figure 4 biomolecules-15-00891-f004:**
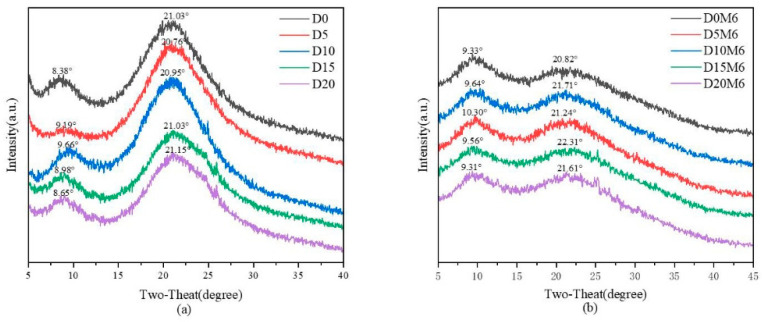
XRD spectra of gelatin/zein nanofibers loaded with 0%, 5%, 10%, 15%, and 20% dihydromyricetin (**a**) before and (**b**) after Maillard reaction for 6 h.

**Figure 5 biomolecules-15-00891-f005:**
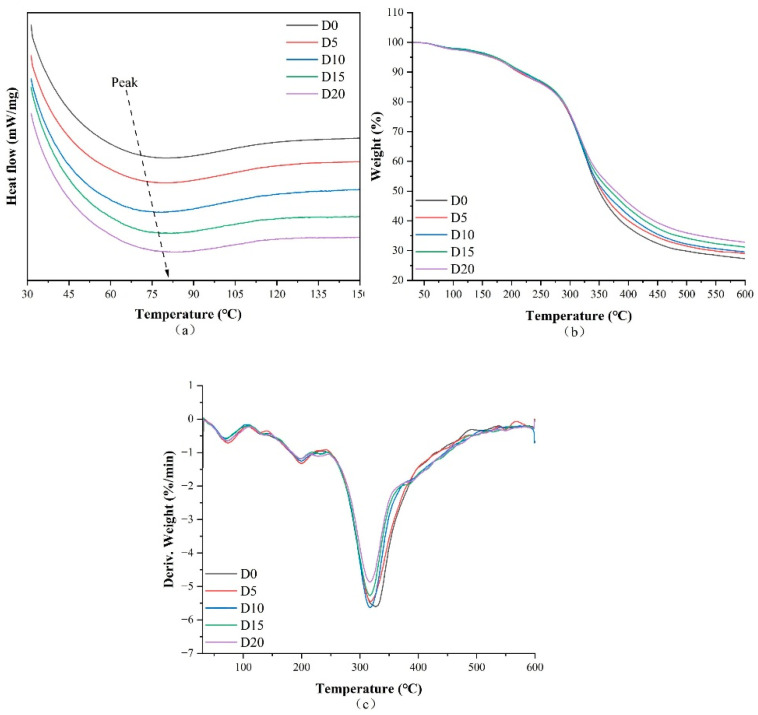
(**a**) DSC, (**b**) TGA, and (**c**) DTG curves of gelatin/zein nanofibers loaded with 0%, 5%, 10%, 15%, and 20% dihydromyricetin.

**Figure 6 biomolecules-15-00891-f006:**
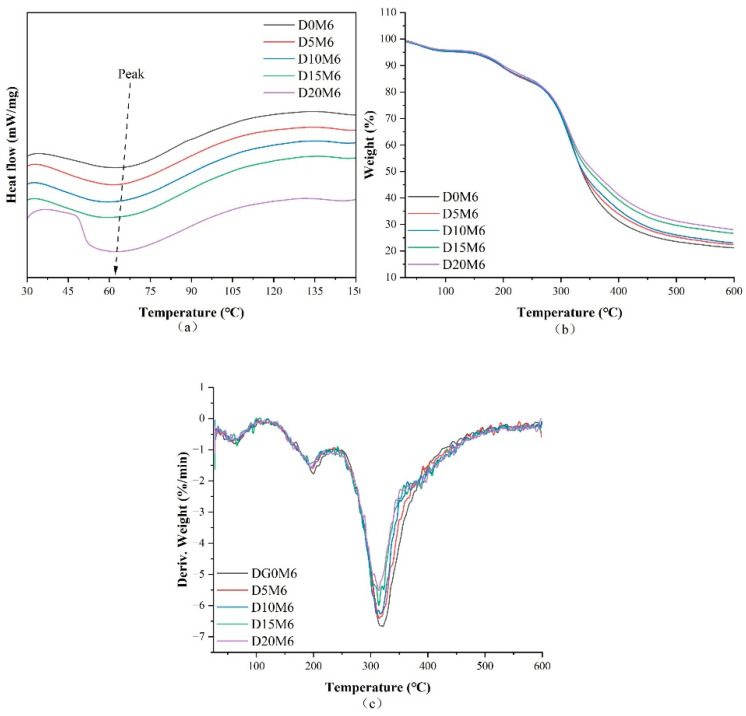
(**a**) DSC, (**b**) TGA, and (**c**) DTG curves of gelatin/zein nanofibers loaded with 0%, 5%, 10%, 15%, and 20% dihydromyricetin after undergoing the Maillard reaction for 6 h.

**Figure 7 biomolecules-15-00891-f007:**
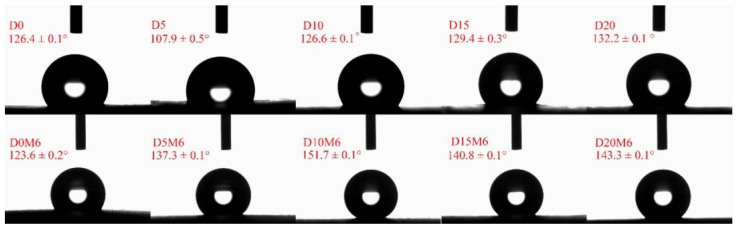
Water contact angles of gelatin/zein nanofibers loaded with 0%, 5%, 10%, 15%, and 20% dihydromyricetin before and after undergoing the Maillard reaction for 6 h.

**Figure 8 biomolecules-15-00891-f008:**
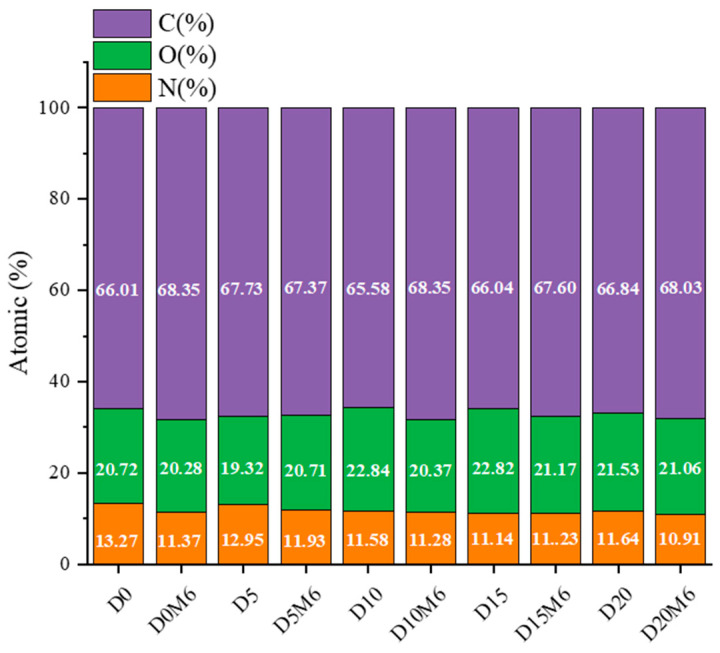
Elemental compositions (atomic%) of gelatin/zein nanofibers loaded with 0%, 5%, 10%, 15%, and 20% dihydromyricetin before and after undergoing the Maillard reaction for 6 h.

**Figure 9 biomolecules-15-00891-f009:**
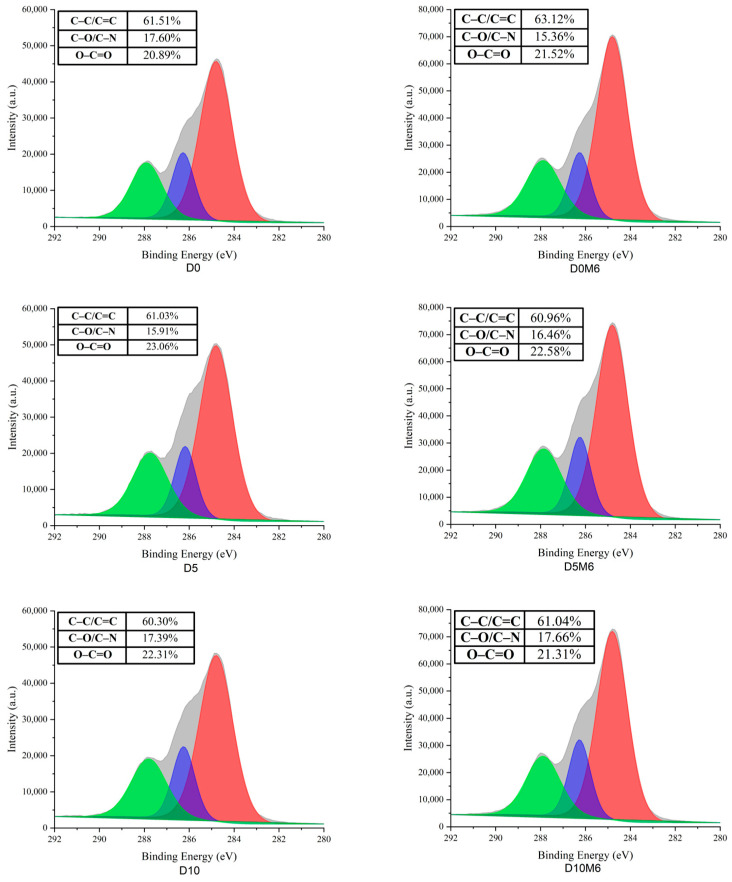
High-resolution C1s XPS spectra of gelatin/zein nanofibers loaded with 0%, 5%, 10%, 15%, and 20% dihydromyricetin before and after undergoing the Maillard reaction for 6 h.

**Figure 10 biomolecules-15-00891-f010:**
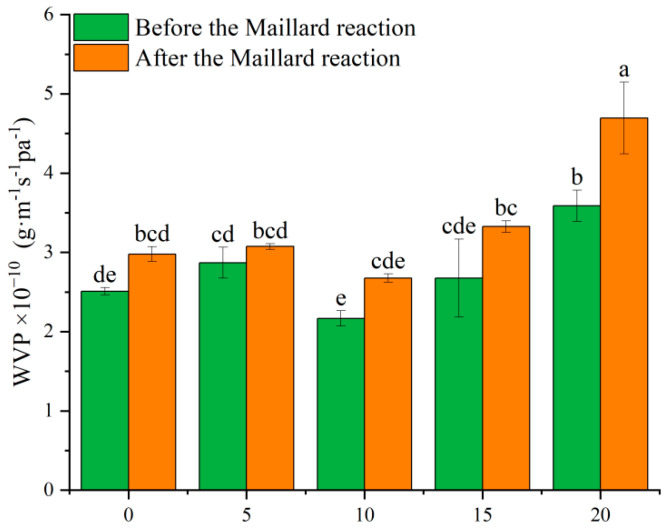
Water vapor permeability of gelatin/zein nanofibers loaded with 0%, 5%, 10%, 15%, and 20% dihydromyricetin before and after undergoing the Maillard reaction for 6 h. Data not sharing the same letter are significantly different between samples (*p* < 0.05).

**Figure 11 biomolecules-15-00891-f011:**
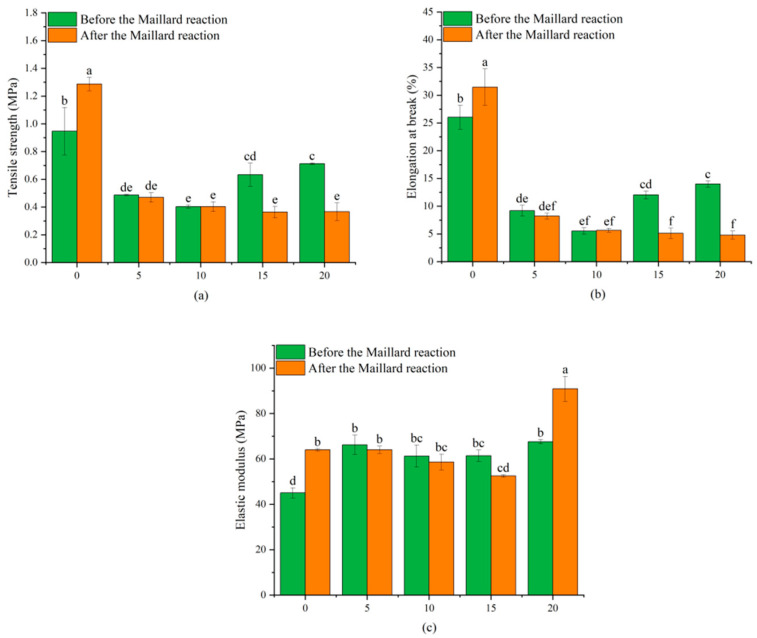
Mechanical properties of gelatin/zein nanofibers loaded with 0%, 5%, 10%, 15%, and 20% dihydromyricetin before and after undergoing the Maillard reaction for 6 h: (**a**) tensile strength; (**b**) elongation at break; and (**c**) elastic modulus. Data not sharing the same letter are significantly different between samples (*p* < 0.05).

**Figure 12 biomolecules-15-00891-f012:**
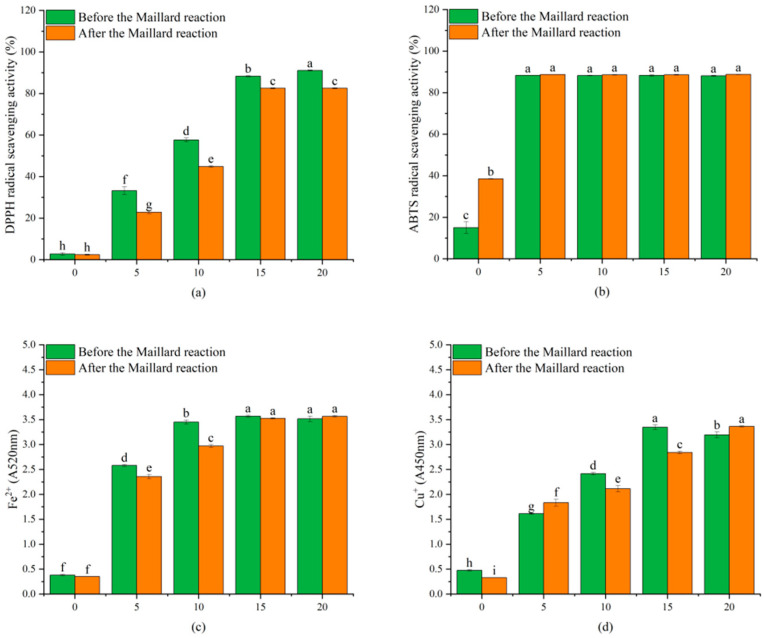
Antioxidant activities of gelatin/zein nanofibers loaded with 0%, 5%, 10%, 15%, and 20% dihydromyricetin before and after undergoing the Maillard reaction for 6 h: (**a**) DPPH; (**b**) ABTS; (**c**) Fe^3+^; and (**d**) Cu^2+^. Data not sharing the same letter are significantly different between samples (*p* < 0.05).

**Figure 13 biomolecules-15-00891-f013:**
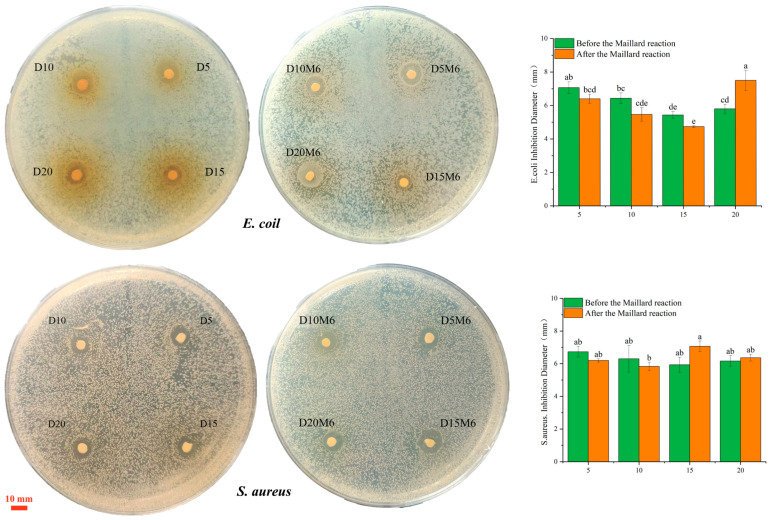
Antibacterial properties of gelatin/zein nanofibers loaded with 0%, 5%, 10%, 15%, and 20% dihydromyricetin before and after undergoing the Maillard reaction for 6 h. Data not sharing the same letter are significantly different between samples (*p* < 0.05).

**Table 1 biomolecules-15-00891-t001:** DSC and TGA data for gelatin/zein nanofibers loaded with 0%, 5%, 10%, 15%, and 20% dihydromyricetin, both before and after the 6 h Maillard reaction.

	DSC		TGA						
	T (°C)	∆H (J/g)	Peak 1 (°C)	Weight Loss (%)	Peak 2 (°C)	Weight Loss (%)	Peak 3 (°C)	Weight Loss (%)	Residue at 600 °C (%)
D0	70.7	−41.270	69.7	1.03	199.7	7.14	326.9	64.50	27.31
D5	72.4	−36.468	73.3	2.47	199.8	5.95	318.3	62.60	28.98
D10	78.8	−18.147	70.1	2.09	199.6	5.78	317.7	62.63	29.51
D15	81.8	−19.692	69.3	2.37	196.0	5.05	317.4	61.39	31.20
D20	82.4	−23.924	73.0	2.67	199.7	5.65	317.6	58.87	32.81
D0M6	66.7	−4.594	61.3	5.16	199.8	11.26	321.1	62.41	21.17
D5M6	64.2	−5.638	58.4	4.86	198.4	10.29	314.8	62.50	22.35
D10M6	64.1	−6.086	57.8	4.82	195.2	10.53	310.1	61.61	23.04
D15M6	65.6	−5.963	66.1	4.67	193.5	10.30	314.4	58.38	26.65
D20M6	63.9	−8.387	51.6	4.47	195.4	11.33	314.0	56.06	28.14

Note: values within the same column accompanied by different superscripts indicate statistically significant differences (*p* < 0.05).

## Data Availability

Data are contained within the article.

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
