# Peer review of "Influence of the Maillard Reaction on the Properties of Gelatin/Zein Nanofibers Loaded with Dihydromyricetin Prepared by Electro-Blowing Spinning"

_biomolecules, 2025, doi:10.3390/biom15060891_

Round 1
Reviewer 1 Report
Comments and Suggestions for Authors
The paper contains a study of gelatin/zein nanofibers loaded with dihydromyricetin with and without Maillard reaction. The fibers were obtained by electro blowing spinning. The characterization was conducted with several techniques like Fourier transform infrared (FTIR) spectroscopy, X-ray diffraction (XRD), mechanical properties, antioxidants and antibacterial properties. Zein/gelatin and dihydromyricetin showed a good compatibility, and the thermal stability improved after Milliard reaction. Also, it is envisaged that the new crosslinked copolymer may have an application in food active packaging.
In point 2.3, the air flow seems to me a bit high to flow in a tiny needle (400 L/h), Could you check this, please?
In the same 2.3 section. Could you improve the details of the scheme? I cannot see who is pushing the syringe plunger, and how the air and the polymer solution get together to flow forward.
Please, check if all acronyms have a definition along the paper, like DPPH, ABTS, TPTZ, ANOVA, PVP, etc.
In the last paragraph of the 2.5 section the definition for WVP appears the second time it was mentioned.
Please, check the subscripts A1 and A0 in the text for the two equations in the section Antioxidant and antimicrobial activities. Also, should this section have a number? Let say, 2.6.
Do you consider the shifts in the 3.2 section are big enough to be considered as part of the discussion? In example:
However, the amide I and amide III shifted from 1647 to 1648 cm−1 and
1247 to 1246 cm−1, respectively.
Section 3.4 XRD spectra analysis
Figure 6. In the figure caption of Figure 10, include a) and b), as you have two graphs in this figure.
Section 3.5 water contact angle
At the last part of this section:
Maillard reaction took place between the carbonyl groups in glucose… Do you mean hydroxyl groups in glucose?
3.10. Antibacterial properties
Your statement:
The inhibitory effect on E. coli was more significant than on S. aureus..
Is that correct if you consider the averages of inhibitory effect of both bacteria?
Figure 13. I suggest you include the results for the 0% content of dihydromyricetin.
Increase the size of the numbers and letters in Figures 3, 9 and 13, please.
Comments on the Quality of English Language3.4 Section, check: van der Waals forces
Please, check whether this sentence is correct:
The crystallinity of the nanofibers had reduced might be the secondary structure of protein had damaged.
2.4. Maillard reaction
The Maillard reaction was carried in
The Maillard reaction was carried out in?
Author Response
1. In point 2.3, the air flow seems to me a bit high to flow in a tiny needle (400 L/h), Could you check this, please?
Response: We appreciate the reviewer’s careful consideration of our methodology. Regarding the airflow rate of 400 L/h in the needle, we acknowledge that this valuemay seem high at first glance. Comprehensively consider the properties of thesolution (such as concentration and viscosity) and their impact on the fiber formationprocess to optimize the experimental parameters. At the same time, upon revisiting
our experimental design and reviewing relevant literature, we confirmthat this flowrate was intentionally selected to meet the specific requirements of our study. Ren, Y.; An, J.; Tian, C.; Shang, L.; Tao, Y.; Deng, L. Tunable Physical Properties of
Electro-Blown Spinning Dextran/Zein Nanofibers Cross-Linked by Maillard Reaction. Foods 2024, 13, 2040. https://doi.org/10.3390/foods13132040
2. In the same 2.3 section. Could you improve the details of the scheme? I cannot seewho is pushing the syringe plunger, and how the air and the polymer solution get
together to flow forward.
Response: Thank you for pointing this issue. We use an automated device to drive thesyringe plunger, and we have included a simplified schematic of this automatedsystem in the illustration to enhance reader comprehension. In the revised manuscript, we have provided a detailed description of this process in Line 125. Regarding the convergence and forward flow of air and the polymer solution, theprocess can be envisioned as follow: the syringe tip is connected to a mixing chamber. Initially, the polymer solution is drawn into the syringe, and then air is introducedintoa specific location within the mixing chamber via a narrow tube, utilizing a pressuredifferential to facilitate preliminary mixing of the two phases. When the plunger is
actuated, the mixture in the chamber is compressed and expelled through the fineorifice at the syringe tip. The increased flow velocity at the outlet further enhances
mixing and ensures continuous forward flow.
3. Please, check if all acronyms have a definition along the paper, like DPPH, ABTS, TPTZ, ANOVA, PVP, etc.
Response: Thank you for pointing this out. The abbreviations have been added in Line104-106, 229, 244.
4. In the last paragraph of the 2.5 section the definition for WVP appears the secondtime it was mentioned.
Response: We sincerely appreciate the reviewer’s thorough review of our manuscript. We acknowledge that the term “water vapor permeability (WVP)” was inadvertently
redefined the final paragraph of Section 2.5, and we have now corrected this in Line
162-166.
5. Please, check the subscripts A1 and A0 in the text for the two equations in the
section Antioxidant and antimicrobial activities. Also, should this section have a
number? Let say, 2.6.
Response: We sincerely appreciate the reviewer’s careful reading. We have checkedthe subscripts A1 and A0 in the section Antioxidant and antimicrobial activities. Andwe have now corrected this in Line 189, 199. Thank you for pointing out the numbering issue. We agree with the reviewer’s
suggestion and have assigned the section number “2.6” to the “Antioxidant and
antimicrobial activities” section to maintain consistent numbering throughout the
manuscript. The number have been added in Line 180.
6. Do you consider the shifts in the 3.2 section are big enough to be considered as
part of the discussion? In example: However, the amide I and amide III shifted from1647 to 1648 cm−1 and1247 to 1246 cm−1
, respectively.
Response: We agree with this comment. Within the margin of error, a shift of 1 cm-1
is extremely minor and can essentially be considered unchanged. We have made the corresponding revisions in Line 277-284.
7. Section 3.4 XRD spectra analysis. Figure 6. In the figure caption of Figure 6, include a) and b), as you have two graphs in this figure.
Response: We sincerely appreciate the reviewer’s careful reading. We have revisedthecaption of Figure 6 (now labeled as Figure 4 in Line 317-318 ) to explicitly indicate
sub-figures (a) and (b) as suggested and the caption of Figure 10 (in Line 421-422). The modified caption now reads:
Figure 4. XRD spectra of gelatin/zein nanofibers loaded with 0%, 5%, 10%, 15%and20% dihydromyricetin (a) before and (b)after Maillard reaction for 6 h. Figure 10. Water vapor permeability of gelatin/zein nanofibers loaded with 0%, 5%, 10%, 15%, and 20% dihydromyricetin before and after the Maillard reaction for 6h.
8. Section 3.5 water contact angle, At the last part of this section: Maillard reactiontook place between the carbonyl groups in glucose… Do you mean hydroxyl groups
in glucose?
Response: We sincerely appreciate the reviewer’s careful reading. The initial reactivegroup in glucose for the Maillard reaction is the aldehyde group (-CHOat C1). The
original text has been modified in Line 370.
9. 3.10. Antibacterial properties, your statement: The inhibitory effect on E. coli was
more significant than on S. aureus..Is that correct if you consider the averages of
inhibitory effect of both bacteria?
Response: Thank you for pointing this important issue. The original text has beenmodified in Line 474-475.
10. Figure 13. I suggest you include the results for the 0% content of
dihydromyricetin.
Response: The antibacterial zone effects of gelatin/zein nanofibers without
dihydromyricetin on Escherichia coli and Staphylococcus aureus were as follows (all
tests were performed in triplicate), showing no significant inhibitory effect, andthus
were not included in the main text. Additionally, Wang et al. also confirmedthat
gelatin/zein nanofibers without loaded bioactive components do not exhibit
significant antibacterial activity. Wang, D.; Liu, Y.; Sun, J.; Sun, Z.; Liu, F.; Du, L.; Wang, D. FabricationandCharacterization of Gelatin/Zein Nanofiber Films Loading Perillaldehyde for thePreservation of Chilled Chicken. Foods 2021,10, 1277. https://doi.org/10.3390/foods10061277
11. Increase the size of the numbers and letters in Figures 3, 9 and 13, please.
Response: Thank you pointing this out. We have made revisions in the manuscript inLine 285, 401, 488.
12. 2.4. Maillard reaction,The Maillard reaction was carried in and The Maillardreaction was carried out in?
Response: We sincerely appreciate the reviewer’s careful reading. Upon careful
consideration, we found “The Maillard reaction was carried in” this expression
inappropriate and have it with “The Maillard reaction was performed in ” in Line 128.
13. Section 3.4, check: van der Waals forces.
Response: Thank you for pointing this out. We have revised “ van der Waals forces ”to “ van der Waals ” in Line 309.
14. Please, check whether this sentence is correct: The crystallinity of the nanofibers
had reduced might be the secondary structure of protein had damaged.
Response: Thank you for your valuable comment. We agree that the original sentencewas unclear. We have revised it to “ The reduced crystallinity of the nanofibers maytherefore be due to damage to the protein secondary structure. ” in Line 311-312. This
change better explains the potential relationship between the reduced crystallizationand protein structure damage.

Reviewer 2 Report
Comments and Suggestions for Authors
Hui Xiang et al. in the article entitled “Influence of the Maillard reaction on the properties of gelatin/zein nanofibers loaded with dihydromyticetin prepared by electro-blowing spinning” investigated a nanocomposite consisted of gelatin/zein nanofibers modified with dihydromyricetin with potential application in active packaging application.
The text is generally well-written and clear, but there are several minor grammatical mistakes, awkward phrasings, and typographic issues. Some sentences are overly long and could benefit from splitting for clarity. There are inconsistencies in tense (past vs. present), article usage, and pluralization. Occasional typographic errors (e.g., misplaced hyphens, inconsistent spacing, and formatting).
Many abbreviations are not listed in the text, e.g. DPPH, ABTS, TPTZ.
The origin of many chemicals is not stated.
While most methods are well described, some experimental details could be further clarified (e.g., exact conditions for some assays, SEM measurements, rationale for certain parameter choices).
I recommend rounding the fiber thickness to whole nm units; Figure 2 – the scale bar is not easy to read.
Comments on the Quality of English LanguageThe text is generally well-written and clear, but there are several minor grammatical mistakes, awkward phrasings, and typographic issues. Some sentences are overly long and could benefit from splitting for clarity. There are inconsistencies in tense (past vs. present), article usage, and pluralization. Occasional typographic errors (e.g., misplaced hyphens, inconsistent spacing, and formatting).
Author Response
Comments:
1. The text is generally well-written and clear, but there are several minor
grammatical mistakes, awkward phrasings, and typographic issues. Some sentences
are overly long and could benefit from splitting for clarity. There are inconsistencies
in tense (past vs. present), article usage, and pluralization. Occasional typographicerrors (e.g., misplaced hyphens, inconsistent spacing, and formatting).
Response: Thank you for pointing this out. we have invited native English speakers torefine the language. And then, we have comprehensively examined the entire text, corrected all the identified grammatical issues (such as subject-predicate agreement, misuse of tenses, article usage, and pluralization, etc), and added conjunctions insentences prone to ambiguity. The typographic errors have been corrected.
2. Many abbreviations are not listed in the text, e.g. DPPH, ABTS, TPTZ.
Response: Thank you for pointing this out. The abbreviations have been added in Line104-106.
3. The origin of many chemicals is not stated.
Response: Thank you for pointing this important issue. We have supplementedandimproved the source information of all the chemical reagents in Line 106-107.
4. While most methods are well described, some experimental details could be further
clarified (e.g., exact conditions for some assays, SEM measurements, rationale for
certain parameter choices).
Response: We agree with your perspective. Having thoroughly reviewedthemethodology section; we have additionally incorporated the relevant parameters for
SEM to enhance the technical rigor of the description in Line 135-137.
5. I recommend rounding the fiber thickness to whole nm units; Figure 2 – the scalebar is not easy to read.
Response: We fully agree with your suggestion and have rounded all fiber thickness
values to the nearest integer nanometer (e.g., D0 “691.8 ± 132.5 nm” has been revisedto “692 ± 133 nm”) to improve data clarity and readability. Additionally, we haveadjusted the scale bar in Figure 2 and replaced it in the revised manuscript
Reviewer 3 Report
Comments and Suggestions for Authors
In the introduction the authors should describe in more detail the areas of application of the materials and the requirements for them, in particular, where the materials will be used and why an increased concentration of dihydromyricetin is necessary, why these polymers were chosen? Why was the blowing process chosen, what advantages does it provide in this case? Why is the Maillard reaction with glucose needed? And provide literature data on the state of research in this area
Can the increased strength with the incorporation of dihydromyricetin also be explained by an increase in the fiber thickness?
The work would benefit from providing a profile of the release of dihydromyricetin from the fiber
When describing the IR, a shift of 1 cm-1 is extremely small within the error limits; it seems impossible to draw conclusions about the shift as a result of the reaction here
Author Response
Comments:
-
Comments:
- In the introduction the authors should describe in more detail the areas of application of the materials and the requirements for them, in particular, Where the materials will be used and why an increased concentration of dihydromyricetin is necessary?
Response: Thank you for pointing out.Nanofibers infused with bioactive compounds like dihydromyricetin are primarily utilized in active food packaging applications, where they effectively provide antimicrobial and antioxidant benefits. This innovative approach enhances the preservation and safety of food products.
Although dihydromyricetin faces challenges such as poor aqueous solubility and limited stability, including susceptibility to photodegradation and thermal decomposition, these issues can be effectively addressed through nanoencapsulation within nanofiber matrices. This encapsulation technique enhances the stability and solubility of dihydromyricetin, thereby expanding its potential applications in food packaging and functional food delivery systems. However, the current low drug loading capacity does not meet the demands for long-term sustained release or high-dose therapeutic applications. Therefore, achieving high-concentration encapsulation is essential to fulfill these objectives, thereby enhancing the therapeutic efficacy and bioavailability of dihydromyricetin in various applications. We have supplemented it in Line 44-47, Line 51-52.
- Why these polymers were chosen?
Response: Thank you for pointing out. Zein, a plant-derived protein, demonstrates excellent biodegradability, biocompatibility, and edibility, while notably enhancing the hydrophobic properties of composite films. Gelatin, a versatile biopolymer derived from collagen, offers excellent biodegradability, emulsification capacity, biocompatibility, and high viscosity, making it highly applicable in the food industry. We have supplemented it in Line 68-72.
- Why werethe blowing process chosen, what advantages does it provide in this case?
Response: Thank you for pointing out. Electro-blowing spinning is a hybrid manufacturing technique that synergistically combines the powerful forces of an electric field with high-pressure gas within a single, efficient apparatus. This innovative method addresses the inherent limitations of traditional processes, such as the low productivity associated with electrospinning and the jet instability commonly observed in solution blow spinning. We have supplemented it in Line 60-62.
- Why is the Maillard reaction with glucose needed?And provide literature data on the state of research in this area.
Response: Thank you for pointing out. In earlier investigations, we successfully synthesized gelatin/zein nanofibers that underwent crosslinking at a temperature of 60°C and a relative humidity of 50%. Following crosslinking with glucose, the fibers demonstrated significantly enhanced water resistance and the formation of a more rigid and reinforced network structure. We have supplemented it in Line 87-91.
- Can the increased strength with the incorporation of dihydromyricetin also be explained by an increase in the fiber thickness?
Response: Thank you for pointing out. Through further discussion, we confirmed that the observed effects are indeed related to changes in fiber diameter. The incorporation of dihydromyricetin increased the solution viscosity, leading to larger fiber diameters and enhanced mechanical strength. Meanwhile, our data suggest that dihydromyricetin reduces the membrane's elasticity, likely due to altered polymer chain interactions. We have revised it in Line 449-450.
- The work would benefit from providing a profile of the release of dihydromyricetin from the fiber.
Response: We agree with this comment. We plan to conduct further experiments on dihydromyricetin in the subsequent stages of the project. However, due to the unavailability of the laboratory's HPLC equipment, we are unable to include the related data in this paper.
- When describing the IR, a shift of 1 cm-1is extremely small within the error limits; it seems impossible to draw conclusions about the shift as a result of the reaction here.
Response: We agree with this comment. Within the margin of error, a shift of 1 cm-1 is extremely minor and can essentially be considered unchanged. We have made the corresponding revisions in Line 287-294.
Reviewer 4 Report
Comments and Suggestions for Authors
The authors studied the preparation and characterization of gelatin/zein nanofiber composites loaded with dihydromyricetin, and their modification using the Maillard reaction. The composites were extensively characterized using different tools and experiments to evaluate the effect of Maillard reaction on the composites and explore their potential application, especially for active packaging.
While the manuscript's topic is interesting and the experimental design is sound, I recommend publication after revisions that address the following:
Given that the main idea is about the effect of the Maillard reaction on the prepared composites, it would be beneficial to include relevant equations and a more detailed explanation of the Maillard reaction and the resulting changes. Introducing this information at the beginning of the Results and Discussion section would significantly enhance reader comprehension. This would provide a clearer framework for understanding the subsequent discussion of FTIR, XRD, TGA, and other characterization results. Please also explicitly connect the chemical structural changes resulting from the Maillard reaction to the observed changes in the various characterization data.
Author Response
Comments 1: Given that the main idea is about the effect of the Maillard reaction on the prepared composites, it would be beneficial to include relevant equations and a more detailed explanation of the Maillard reaction and the resulting changes. Introducing this information at the beginning of the Results and Discussion section would significantly enhance reader comprehension. This would provide a clearer framework for understanding the subsequent discussion of FTIR, XRD, TGA, and other characterization results. Please also explicitly connect the chemical structural changes resulting from the Maillard reaction to the observed changes in the various characterization data.”
Response: Thank you for pointing this out. The Maillard reaction is an extremely complex process, and although it is divided into three stages, it can undergo numerous variations. The reaction process on our nanofiber membranes differs from that in conventional solution systems. The primary objective of our study is to investigate the changes in the properties of nanofiber membranes after the Maillard reaction, exploring whether the Maillard reaction can serve as a method to modify nanofibers. Once again, we sincerely appreciate the reviewer's insightful question—it is an excellent suggestion. In our follow-up research, we will conduct detailed analyses of the reaction products and their correlations with macroscopic properties.
Round 2
Reviewer 4 Report
Comments and Suggestions for Authors
The authors answered the reviewer's questions.